# Real-Time Evaluation of Optic Nerve Sheath Diameter (ONSD) in Awake, Spontaneously Breathing Patients

**DOI:** 10.3390/jcm10163549

**Published:** 2021-08-12

**Authors:** Nick Weidner, Jessica Kretschmann, Hagen Bomberg, Sebastian Antes, Steffen Leonhardt, Christoph Tschan, Joachim Oertel, Thomas Volk, Andreas Meiser, Heinrich V. Groesdonk

**Affiliations:** 1Department of Anesthesiology, Intensive Care Medicine and Pain Medicine, Saarland University Medical Center, 66421 Homburg, Germany; nick.weidner@helios-gesundheit.de (N.W.); jessica.kretschmann@uks.eu (J.K.); hagen.bomberg@uks.eu (H.B.); thomas.volk@uks.eu (T.V.); andreas.meiser@uks.eu (A.M.); 2Department of Neurosurgery, Saarland University Medical Center, 66421 Homburg, Germany; sebastian.antes@uks.eu (S.A.); christoph.tschan@uks.eu (C.T.); joachim.oertel@uks.eu (J.O.); 3Helmholtz Institute for Biomedical Engineering, RWTH Aachen University, 52074 Aachen, Germany; medit@hia.rwth-aachen.de

**Keywords:** elevated intracranial pressure, non-invasive ICP measurement, ONSD

## Abstract

(1) Background: Reliable ultrasonographic measurements of optic nerve sheath diameter (ONSD) to detect increased intracerebral pressure (ICP) has not been established in awake patients with continuous invasive ICP monitoring. Therefore, in this study, we included fully awake patients with and without raised ICP and correlated ONSD with continuously measured ICP values. (2) Methods: In a prospective study, intracranial pressure (ICP) was continuously measured in 25 patients with an intraparenchymatic P-tel probe. Ultrasonic measurements were carried out three times for each optic nerve in vertical and horizontal directions. ONSD measurements and ICP were correlated. Patients with ICP of 2.0–10.0 mmHg were compared with patients suffering from an ICP of 10.1–24.2 mmHg. (3) Results: In all patients, the ONSD vertical and horizontal measurement for both eyes correlated well with the ICP (Pearson R = 0.68–0.80). Both measurements yielded similar results (Bland-Altman: vertical bias: −0.09 mm, accuracy: ±0.66 mm; horizontal bias: −0.06 mm, accuracy: ±0.48 mm). For patients with an ICP of 2.0–10.0 mmHg compared to an ICP of 10.1–24.2, ROC (receiver operating characteristic) analyses showed that ONSD measurement accurately predicts elevated ICP (optimal cut-off value 5.05 mm, AUC of 0.91, sensitivity 92% and specificity 90%, *p* < 0.001). (4) Conclusions: Ultrasonographic measurement of ONSD in awake, spontaneously breathing patients provides a valuable method to evaluate patients with suspected increased ICP. Additionally, it provides a potential tool for rapid assessment of ICP at the bedside and to identify patients at risk for a poor neurological outcome.

## 1. Introduction

The elevation of intracranial pressure (ICP), defined as ICP > 20 mmHg, is a common life-threatening condition caused by a variety of traumatic and non-traumatic diseases. Untreated intracranial hypertension can lead to severe brain damage with a poor neurologic outcome or patients’ death due to secondary ischemia or brainstem herniation, respectively [1,2].

Invasive intracranial pressure monitoring currently is the gold standard to detect intracranial hypertension. Indications for the application of these devices are based on clinical observations. For example, it is recommended to implement invasive monitoring in patients with: (i) severe traumatic brain injury; (ii) multiple injuries with an altered level of consciousness; (iii) a post-resuscitation Glasgow coma score (GCS) of 8 or less after resuscitation in the presence of an abnormal cranial CT scan (cCT); (iv) a normal cCT but >2 risk factors (systolic blood pressure, SBP < 90 mmHg, decorticate or decerebrate posturing); or (v) reduced GCS subsequent to the removal of an intracranial mass [3,4]. In daily clinical praxis, the difficulty lies in patients with GCS between 9 and 12 who may benefit from aggressive medical therapy, which can only be accurately initiated and monitored when ICP is measured invasively. However, this invasive and not always feasible technique can lead to complications such as hemorrhage, malfunction, or infection [3,5], which explains why the decision to undertake these procedures can often be difficult.

Unfortunately, neuroimaging such as cranial computed tomography scan has often limited availability, requires potential harmful patient transport, and has poor performance for detection of raised ICP [6,7]. Additionally, ICP may be highly dynamic—almost instantly changing its value from normal baseline to severely elevated levels [8].

In contrast, point of care ultrasonography of optic nerve sheath diameter (ONSD) has become an alternative bedside tool to reproducibly detect intracranial hypertension [9,10,11,12]. The optic nerve is surrounded by all linings of the brain, forming the optic nerve sheath, which connects it to the intracranial space. Intracranial pressure is conducted through the nerve sheath, which, in the case of elevation of intracranial pressure, widens. In this context, ONSD with more than 5.0 mm was associated with increased ICP in severely traumatic and non-traumatic injured patients [13,14,15]. Additionally, human studies have shown that widening of ONSD occurs within minutes after acute changes in ICP [11,12,16,17,18].

## 2. Materials and Methods

After approval by the local ethics committee (Landesärztekammer des Saarlandes; Ref. ID: 151/13), this study was conducted from January 2014 to March 2015 at the Saarland University Medical Center in Homburg, Germany. It was designed as a prospective observation trial, and written informed consent was obtained from all 25 patients included in this study.

### 2.1. Patient Cohort

Patients included in this study had a continuous ICP measuring device implanted (Neurovent P-tel, Raumedic AG, Helmbrechts, Germany) due to suspected or proven intracranial hypertension. All patients were free of major symptoms of elevated intracranial pressure and hospitalized under normal ward conditions. None of the patients included were admitted under emergency conditions or showed the urge for immediate treatment.

The probe was implanted through a precoronal and parasagittal burr hole (at Kocher’s point). After the burr hole was drilled, dura and pia mater were incised cruciform. The polyurethane catheter of the probe was carefully advanced through the parenchyma of the frontal lobe. The housing of the telemetric device remained above the skull surface. All probes were implanted in the Department of Neurosurgery.

Measurements of ICP can be started directly after implantation by placing the TDTreadP (TDTreadP, Raumedic AG, Helmbrechts, Germany), a special reading unit, above the closed wound. All data were saved on a datalogger (Datalogger MPR 1, Raumedic AG, Helmbrechts, Germany). Data readout was attained with special software Datalog (Datalog, Raumedic AG, Helmbrechts, Germany).

### 2.2. ONSD Measurement

The reading unit (TDTreadP) was placed and secured over the implanted P-Tel probe at least 24 h prior to the ONSD measurement procedure. The reading unit remained above the P-Tel probe until ONSD measurements were finished. While the ONSD measurement was done, ICP values were not visible to the examining physician.

All measurements were carried out in the afternoon to avoid bias by circadian rhythm changes. For the measurement, patients rested in the supine position for five minutes. Meanwhile, the presence of increased intracranial pressure was checked by excluding typical symptoms. When patients stayed without clinical signs of exacerbating ICP, measurement of OSND was performed. ICP Data were stored by the datalogger for every second, and a readout of total data was carried out afterwards. For statistical analysis, we used mean values of recorded ICP data for each measurement.

The optic nerve sheath diameter was measured using a LogicE^®^ system (LogicE^®^, Fa. GE Healthcare, Solingen, Germany) with a high resolution (7–10 MHz) linear array ultrasound transducer probe (9L-RS). Penetration-depth was optimized to generate an image of the eye filling the screen. Preset “small parts”, usually used for an ultrasound of nerval structures, provided as a standard ultrasound preset by the device was chosen to avoid exceeding tissue and mechanical index. To ensure an energy level not exceeding current recommendations, we took care of mechanical and tissue index (MI < 0.2; TI < 1.0) while adjusting emitted energy of the ultrasound probe [19,20]. A large amount of standard water-soluble ultrasound transmission gel was applied to the patient’s closed eyelid (Aquasonic100^®^, Fa. C+V Pharma Depot GmbH, Versmold, Germany).

Before starting the measurement, the awake patient was placed in the supine position and left for 5 min. Within this time, cardiopulmonary monitoring was installed.

In accordance with current recommendations, the globe was scanned in the transverse and horizontal plane in both eyes and the ONSD was determined at a predefined point 3 mm posterior to the globe [21]. ONSD was measured three times transversally and horizontally directly after each other. Additionally, non-invasive blood pressure, heart rate and peripheral oxygen saturation were collected during the measurement.

### 2.3. Statistical Analysis

All acquired data were anonymized and transferred to our study database (Microsoft^®^ Excel 2010, Microsoft^®^ Corporation, Redmond, WA, USA). All data were verified for integrity and plausibility on the day of the patient’s discharge.

Fisher’s exact test was performed to compare patients with lower intracranial pressure (≤10 mmHg) and higher intracranial pressure (>10 mmHg) regarding dichotomous variables. For continuous variables, the differences between the two groups were compared using Student’s *t*-tests (Welch’s *t*-tests in case of inhomogeneous variances, respectively). Continuous variables were expressed as mean and standard deviation (SD). Two-sided *p*-values of less than 0.05 were considered statistically significant.

A Bland Altman diagram was used to compare the measurement of the ONSD horizontal eye and the ONSD measurement of the vertical eye. Accuracy and precision were calculated for horizontal and vertical measurements. Linearity between the ONSD and the P-tel probe measurement was tested by Pearson correlation coefficients. Different receiver operating characteristic (ROC) curves were constructed to evaluate the predictive power of horizontal and vertical ONSD measurements for the occurrence of increased intracranial pressure (>10 mmHg). The Youden Index was used to calculate optimal cut-off points for ONSD measurement for the prediction of increased intracranial pressure (>10 mmHg). The positive predictive value is defined as the number of true positives/(number of true positives + number of false positives). The negative predictive value is defined as the number of true negatives/(number of true negatives + number of false negatives). All data analyses were performed using SPSS Statistics 19™ (IBM, Ehningen, Germany).

## 3. Results

Patient’s characteristics are shown in Table 1. Among these two groups, no statistical differences were found.

### 3.1. Correlation of the ONSD with Intracranial Pressure

Mean values of each three horizontal and vertical ONSD measurements of the left eye correlated well with the ICP measured by the P-tel probe (horizontal: Pearson R = 0.74; vertical: Pearson R = 0.68; Figure 1). Also, the corresponding mean values of ONSD measurements of the right eye correlated well with the ICP (horizontal: Pearson R = 0.79; vertical: Pearson R = 0.80; Figure 2).

### 3.2. Accuracy and Precision of ONSD Measurements

ONSD of the left eye measured by the horizontal view conformed to that of the vertical view with high accuracy (bias: −0.09 mm) and precision (two standard deviations of measurement error: ±0.66 mm, Figure 3).

ONSD of the right eye measured by the horizontal view conformed to that of the vertical view with high accuracy (bias: −0.06 mm) and precision (two standard deviations of measurement error: ±0.48 mm, Figure 3).

### 3.3. ROC Analysis

The predictive value of ONSD measurements for the occurrence of high normal range intracranial pressure (>10 mmHg) showed an area under the curve (AUC) of 0.91 (95% CI, 0.83–0.99; *p* < 0.001), a sensitivity of 92%, and a specificity of 90%. The optimal cutoff value for ONSD measurement was found to be 5.1 mm (Figure 4).

The positive predictive value of the ONSD horizontal and vertical measurement of both eyes to identify patients with an increased ICP (10.1–24.2 mmHg) was 0.97; the negative predictive value was 0.73.

## 4. Discussion

In this prospective study, ONSD predicts increased intracranial pressure (>10 mmHg) with high accuracy and precision in awake, spontaneously breathing patients. To the best of our knowledge, this is shown for the first time using an intraparenchymatic P-tel probe which allows real-time measurements of intracranial pressure (ICP) in awake patients. Thus, our results indicate that ultrasonographic measurement of ONSD could be a strong predictor of elevated ICP in this patient cohort.

Brain damage is related to a direct “primary” lesion, such as intracerebral bleeding, severe cerebral concussion, and brain tumors, or to indirect “secondary” causes, such as diffuse prolonged cerebral hypoxia. Eventually, these conditions may result in severe dramatic brain edema with uncontrollable intracranial hypertension [22]. Importantly, intracranial hypertension can occur suddenly without early warning signs and may be highly dynamic, thereby changing its value from normal to severely elevated within minutes [8]. This underlines the need for a bedside monitoring tool to detect raised ICP immediately and to initiate adequate therapy as early as possible. This is especially true in spontaneously breathing patients who suffer from brain injury but did not initially receive any invasive ICP monitoring device.

Today, ONSD evaluation with ultrasound is well accepted as a reliable indicator of intracranial hypertension, with high intra- and inter-observer reliability and with a whole range from 4.3 to 7.6 mm [23,24]. In patients suffering from severe brain damage, ONSD values of about 5.9 to 6.3 mm have been reported [14] with no values lower than 5.8 mm in cases where ICP is >20 mmHg [25]. Therefore, a cut-off value of 5 mm has been suggested [9,26,27]. Despite these numerous studies, there is no consensus regarding a definitive threshold for elevated ICP [21,28,29], which is even true for awake, spontaneously breathing patients with cerebral pathologies.

Our findings are in line with previously mentioned results, showing a slightly wider optic nerve sheath diameter in patients with increased ICP, ranging from 3.8 to 5.6 mm, and an optimal cut-off value for the prediction of increased ICP of 5.1 mm. This was true for both eyes, independent of the implantation side of the pressure probe. Of note, we defined increased ICP > 10 mm Hg, pointing out that ultrasonographic measurement of ONSD is a highly reliable tool to detect ICP even in the upper normal range. This is important in the way that ONSD sonography seems to detect patients at risk for developing severe cerebral complications at a very early stage.

Nevertheless, the importance of our results that represent our final take-home message is not to promote a single ONSD measurement or the absolute value of ONSD as an adequate stand-alone monitoring tool in patients with brain injury but to point out the capability of this non-invasive method as a first-line, bedside diagnostic tool that can be easily and repeatedly performed.

### Limitations

Our present cohort consisted only of patients with hydrocephalus. Implantation of P-Tel probe is indicated narrowly, which leads to a low number of total patients and a long study period. We, therefore, deliberately included these patients who received a P-tel probe implant prior to the study, offering the possibility to measure online ICP in awake, spontaneously breathing patients. Nevertheless, the population examined was small and inhomogeneous to strongly support the results.

The collected data of ONSD were correlated with mean values of ICP values of the whole measurement procedure. Therefore, the influence of fluctuating ICP values on statically measured ONSD cannot be ruled out. Nevertheless, we assume that this influence is of less importance because patients rested in the supine position for five minutes before starting ONSD evaluation. Because of this, potentially raised ICP should be recognized by our repeated measurements.

We are aware that the underlying pathology differs from patients with traumatic or ischemic brain injury, but as all these pathologies result in the same final course, we believe that this is of less importance but needs to be kept in mind while interpreting our data.

## 5. Conclusions

Ultrasonographic measurement of ONSD in awake, spontaneously breathing patients provides a valuable method to evaluate patients with suspected increased ICP. As a non-invasive monitoring method, it provides a potential bedside tool for rapid assessment of ICP, even when still in the upper normal range, to identify patients at risk for a poor neurological outcome.

## Figures and Tables

**Figure 1 jcm-10-03549-f001:**
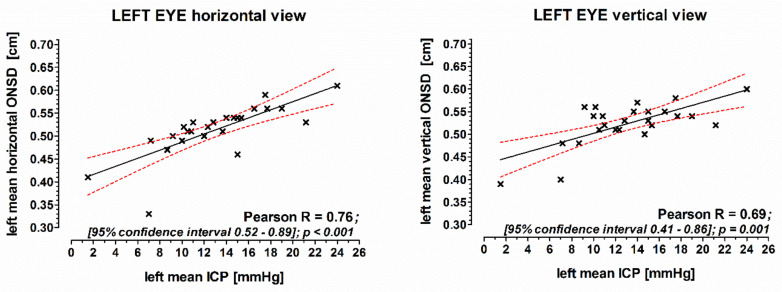
The correlation of the mean of ONSD measurements of the left eye with intracranial pressure. ICP was measured continuously with an intraparenchymatic P-tel probe. ONSD: optic nerve sheath diameter; ICP: intracranial pressure; Pearson: Pearson correlation coefficient.

**Figure 2 jcm-10-03549-f002:**
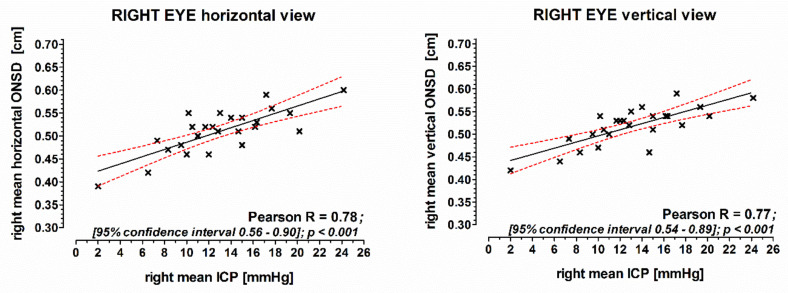
The correlation of the mean of ONSD measurements of the right eye with intracranial pressure. ICP was measured continuously with an intraparenchymatic P-tel probe. ONSD: optic nerve sheath diameter; ICP: intracranial pressure; Pearson: Pearson correlation coefficient.

**Figure 3 jcm-10-03549-f003:**
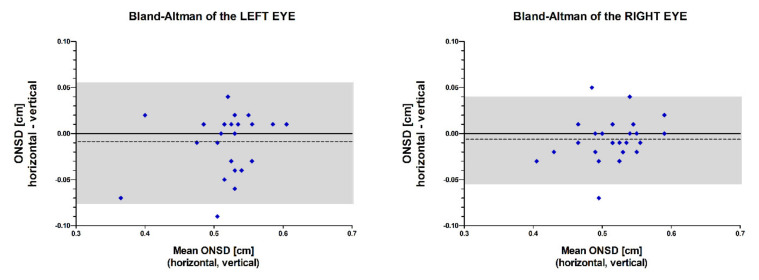
A Bland Altman diagram for comparison of the ONSD (optic nerve sheath diameter) measured by two types of measurements: one with horizontal view and one with vertical view. The difference between the measurements for each ONSD is plotted against the mean of the two measurements. The bias (−0.009 cm, dotted line) and random measurement disagreement (two standard deviations of measurement error: ±0.066 cm, shaded area) are small for the left eye. The bias (−0.006 cm, dotted line) and random measurement disagreement (two standard deviations of measurement error: ±0.048 cm, shaded area) are small for the right eye.

**Figure 4 jcm-10-03549-f004:**
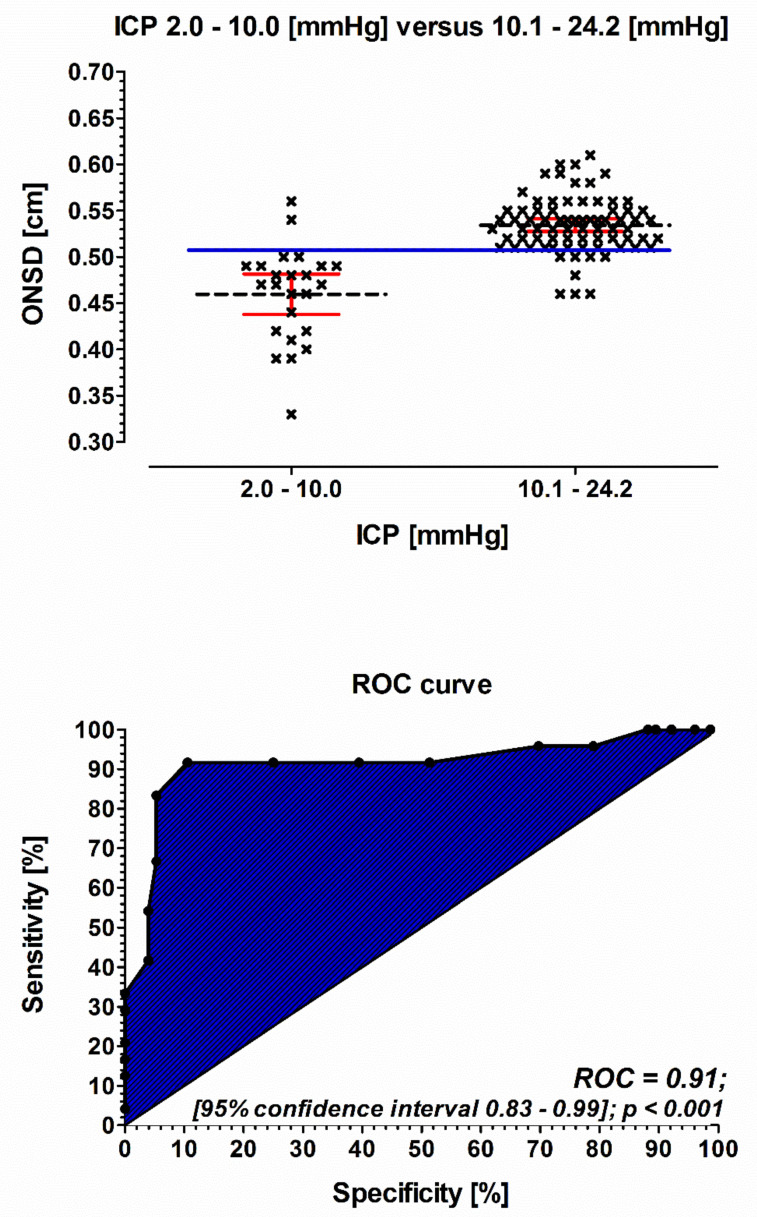
A box plot of ONSD measurement (upper picture). Comparison of ONSD measurements in patients suffering from ICP of 2.0–10.0 mmHg (*n* = 6) versus an ICP of 10.1–24.2 mmHg (*n* = 19). Blue solid line: cut off value 0.51 cm. Black short dash = mean values of the ONSD measurement. Red line: 95% confidence interval. ROC curve analysis of the prognostic accuracy of ONSD measurement (lower picture). ONSD: optic nerve sheath diameter; ICP: intracranial pressure (was measured continuously with an intraparenchymatic P-tel probe).

**Table 1 jcm-10-03549-t001:** Table showing patients characteristics. ICP: intracranial pressure.

	ICP 1.0–10.0 mmHg		ICP 10.1–24.2 mmHg		*p*-Value
	(*n* = 6)		(*n* = 19)		
Male (%)	2	(33)	5	(26)	0.4
Age (mean ± SD)	54	±24	57	±17	0.8
Body mass index (kg/m^2^)	29	±6	30	±8	0.9
Mean arterial pressure (mmHg)	94	±15	98	±18	0.6
Heart rate (beats per minute)	79	±4	77	±14	0.5
Peripheral capillary oxygen saturation (%)	97	±2	96	±2	0.2
Diagnosis (%)					
Hydrocephalus	2	(33)	9	(47)	0.1
Normal pressure hydrocephalus	2	(33)	5	(26)	1
Shunt dysfunction	2	(33)	4	(21)	1

## Data Availability

Not applicable.

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
