# Peer review of "Real-Time Evaluation of Optic Nerve Sheath Diameter (ONSD) in Awake, Spontaneously Breathing Patients"

_jcm, 2021, doi:10.3390/jcm10163549_

Round 1
Reviewer 1 Report
Dear Authors,
I am well aware that the review process is long and often gets in the way of the publication time of a study, but it is itself the reason for the quality guarantee of what is published. It is often a process of confrontation between experts. With my previous observations, you have made few changes to the manuscript, I ask you to continue reading it to understand, from the reader's point of view, what are still the objective difficulties in publishing it. I have recovered your comments on my observations still worthy of further study, adding my new 10 observations in italics.
1] Materials and Methods, first paragraph: the period in which the study was carried out is now fairly dated and the number of patients studied is relatively small to statistically support the conclusions. I am well aware of the difficulties in enrollment these patients, but it is necessary to include these concepts in the Limitations of the manuscript.
We changed lines 213-214 in the limitations section and included it.
as pointed out in my previous comment, from my point of view, it is necessary to add in the limitations a clear concept in the Limitations: the population examined is small and inhomogeneous to strongly support the results.
2] Materials and Methods, 2.1. Patient Cohort: the population cohort enrolled is represented by patients diagnosed with hydrocephalus, normotensive hydrocephalus and with malfunctioning shunt. From my point of view, it is necessary to describe in detail the characteristics of these patients. Are they acute patients? Patients with hydrocephalus, what diagnoses did they have? How do they stay awake in the ward with invasive ICP monitoring? Why are they still deserving of an ICP probe?
3] Materials and Methods, 2.2. ONSD Measurement: about ultrasound setting, which was the acoustic output of the equipment? Unfortunately, even if there are many other papers about ONSD, doesn’t still exist a sheared consensus on ultrasound ONSD setting.
We took presets which came with manufacturer settings. We used preset small parts of GE. We took care of tissue index and mechanical index while ultrasound examination. Guidelines for safety in ultrasound recommend to take care of TI and MI.
We added lines 92 – 94 in Materials and Methods 2.2. to describe the main settings.
Unfortunately, the "small parts" setting does not automatically include the reduction of Acoustic Output. This is a very specific recommendation and an important precaution. We cannot spread the idea that ONSD can be achieved with the presets of a normal ultrasound machine. I ask you to check which acoustic output you have used and to include the FDA reference among the references
Food and Drug Administration. Marketing clearance of diagnostic ultrasound systems and transducers—draft guidance for industry and Food and Drug Administration Staff. 2019. https ://www.fda.gov/media /71100 /download.
4] Materials and Methods, 2.2. ONSD Measurement, line 87: the Authors wrote that they placed the patients supine for 5 minutes before the ONSD measurement: I wonder if this was done indistinctly by the measured ICP, especially for patients with invasive ICP greater than 20 mmHg. For example: which were the CPP values in these patients with intracranial hypertension? Please specify this clinical choice well.
Readout of ICP values was done afterwards and current ICP values during measurement were not visible to examiner. We only examined patients who were admitted to normal care wards. These patients were kept_ under normal conditions doing their “everday day life” on the ward. Being aware of we monitored current state of consciousness and symptoms for elevation of ICP.
We therefor added it into patient cohort description in Materials and Methods lines 71 -72
Dear Authors, thank you for your explanation. Unfortunately, however, I still do not understand how it is possible that a patient with ICP> 20 mmHg can carry out his "every day life", as you describe it. What do you mean by "being aware of ..... state of consciuosness ..."; I understood that the invasive ICP reading was done later, but perhaps there was some altered data or artifacts? For example a patient who coughs? Please, try to make the scenario explainable, either by removing data greater than 20 or by specifying that it is point-like data of insignificant duration.
5] Materials and Methods, 2.2. ONSD Measurement, second paragraph: Intracranial hypertension is widely recognized as a medical emergency. To avoid herniation, head of bed at 30 °, analgesia / sedation, is already foreseen in Tier 0 to minimize metabolic requirements and minimize noxious stimuli ( 10.1007/s12028-017-0454-z ). I cannot imagine the context of at least 5 out of 25 patients with ICP greater than 20 mmHg kept awake in spontaneous breathing. Please specify well the safety context in which the study took place.
We think that this needs no further mentioning, because all of our patients were hospitalized under normal ward conditions. We found elevated ICP after data read out later in time.
This previous observation of mine is also due to the difficulty of understanding those ICP data above 20 mmHg Please specify well the safety context in which the study took place.
6] Materials and Methods, 2.2. ONSD Measurement, third paragraph: the Authors report having performed the measurements of ONSD with ultrasound three times: why this choice? at what distance in time did the measurements take place? Did the statistical analysis take into account the invasive ICP value at the time of the ultrasound measurement, or was the continuous and discontinuous variable compared? Please specify well.
ICP measurement was continuously, ONSD measurements were carried out one after the other. We used three measurements to prevent false measurements and calculated mean values. We also implemented this into clinical procedure to be safe to not have false measurements. We added a short description in line 103.
so if I understand correctly, the invasive ICP data is continuous and collected in 24 hours or more, while the ONSD data is punctual, ie performed in a short space of time. They are therefore measures in different conditions, on the same patients, but at different times of their day. The design of this study must be specified in great detail to make it understandable to the reader.
7] Table 1: 7 patients with normal pressure hydrocephalus are included in the study, of which 5 are included in the 10-24 mmHg group. These patients do not have intracranial hypertension by definition. I ask the Authors if they have measured ICP> 20 mmHg in these patients. The case mix objectively brings together very different pictures of ICP modification: in the limitations the authors argue that this has an insignificant weight. I believe instead that the rapidity of increase of intracranial hypertension has a great influence on ONSD and in such a small sample, the inhomogeneity of patients becomes influential. I would modify the paragraph on Limitations and reformulate the statements in the Discussion / Conclusion / abstract
Literature showing different cut off values for elevated ICP. Usually higher values of ONSD are correlated with ICP above 20mmHg. To our own surprise, we could show that even in upper normal range of ICP there is a detectable cut off value, improving the predictive value of the cut off value.
On normal pressure hydrocephalus, I just can't accept the idea that there are patients with intracranial hypertension. This must be distinguished and specified, even if data was collected after ONSD.
8] 3.ROC Analysis, first paragraph: Is 5.1 the value in millimeters of ONSD corresponding to 10 mmHg of invasive ICP? It seems to me a value quite in contrast with the literature data that define ICP> 20 mmHg with ONSD between 5 and 6 mm. Please specify this ROC result well.
We changed it to high normal range ICP lines 161 - 162
Dear Authors, also the literature you cite intends to consider ONSD cut off values between 5 and 6 mm, for ICP> 20 mmHg, which is what we find in daily clinical practice. It is no surprise that you find an association for values> 10 mmHg, it is probably due to the fact that continuous versus point values are included. This is a bias of the manuscript and must be specified, we cannot pass the concept that 5.1 mm of ONSD corresponds to ICP 10 mmHg. Please rephrase this conclusion and express it correctly.
9] Discussion, line 181: the Authors state citing only one work that US ONSD assessment has "high intra- and inter- observer reliability". It would be very nice if that were the case, but unfortunately we know that this, like most of the ultrasound methods, is very operator dependent. We are looking for new biomarkers / settings to standardize the procedure. Please review the statement.
“”””literature
no answer to my observation has been received. Please integrate
10] References: unfortunately the references should be completely updated, as they all predate the period of the study (2015), which have therefore been partially overtaken by stronger papers, some of which I have already indicated in the review.
no answer to my observation has been received. Please integrate
Author Response
Dear Reviewer,
We have to apologize for the late response. Unfortunately, i had a severe accident which led to a long hospital and rehab stay. Therefor i could not complete the manuscript. We answered all of your advises and included it within the manuscript.
with best regards
Nick Weidner

Reviewer 2 Report
The authors responded sincerely to my questions or requests for corrections, and made appropriate corrections.
I think this is good.
Author Response
Dear Reviewer,
We have to apologize for the late response. Unfortunately, i had a severe accident which led to a long hospital and rehab stay. Therefor i could not complete the manuscript.
with best regards
Nick Weidner
Reviewer 3 Report
No further comments.
My remarks have been addressed in a sufficient manner.
Author Response
Dear Reviewer,
We have to apologize for the late response. Unfortunately, i had a severe accident which led to a long hospital and rehab stay. Therefor i could not complete the manuscript.
with best regards
Nick Weidner
This manuscript is a resubmission of an earlier submission. The following is a list of the peer review reports and author responses from that submission.
Round 1
Reviewer 1 Report
Dear Authors,
It was a great pleasure to read and review your manuscript. Non-invasive ICP monitoring through ONSD is my preferred topic! I hope that my small suggestions can be useful for the publication of this paper. Below, you will find point by point my observations.
- Introduction: in the second paragraph Authors describe punctually the indication for invasive ICP monitoring: I don’t think that this add useful information to the readers about the main topic: the ICP non-invasive ONSD assessment.
- Introduction, pag 2, line 56: Authors wrote “50 mm” describing ONSD value: I am sure this is a typo
- Materials and Methods, first paragraph: the period in which the study was carried out is now fairly dated and the number of patients studied is relatively small to statistically support the conclusions. I am well aware of the difficulties in enrollement these patients, but it is necessary to include these concepts in the Limitations of the manuscript.
- Materials and Methods, 2.1. Patient Cohort: Authors described the invasive ICP monitor technique. Were patients submitted to the procedure in the operating room? Were they in general anesthesia? Which is the time between general anesthesia and the awake ICP assessment. Neurosurgeons used a bolt or surgical drilling? Did the neurosurgeons set to zero correctly in water? Please, specify.
- Materials and Methods, 2.2. ONSD Measurement: about ultrasound setting, which was the acoustic output of the equipment? Unfortunately, even if there are many other papers about ONSD, doesn’t still exist a sheared consensus on ultrasound ONSD setting.
- Materials and Methods, 2.2. ONSD Measurement, line 87: the Authors wrote that they placed the patients supine for 5 minutes before the ONSD measurement: I wonder if this was done indistinctly by the measured ICP, especially for patients with invasive ICP greater than 20 mmHg. For example: which were the CPP values in these patients with intracranial hypertension? Please specify this clinical choice well.
- Materials and Methods, 2.2. ONSD Measurement, second paragraph: Intracranial hypertension is widely recognized as a medical emergency. To avoid herniation, head of bed at 30 °, analgesia / sedation, is already foreseen in Tier 0 to minimize metabolic requirements and minimize noxious stimuli ( 10.1007/s12028-017-0454-z ). I cannot imagine the context of at least 5 out of 25 patients with ICP greater than 20 mmHg kept awake in spontaneous breathing. Please specify well the safety context in which the study took place.
- Materials and Methods, 2.2. ONSD Measurement, third paragraph: the Authors report having performed the measurements of ONSD with ultrasound three times: why this choice? at what distance in time did the measurements take place? Did the statistical analysis take into account the invasive ICP value at the time of the ultrasound measurement, or was the continuous and discontinuous variable compared? Please specify well.
- 3. Statistical Analysis: the Authors they chose 10 mmHg with ICP value to dichotomize patients in ROC. Intracranial pressure, according to the guidelines, reaches critical values at 20 mmHg and requires treatment when it is higher than 22 mmHg ( https://braintrauma.org/uploads/07/04/Guidelines_for_the_Management_of_Severe_Traumatic.97250__2_.pdf ). How was the 10 mmHg value chosen? It is a commonly measurable value even in non-pathological pictures. Please specify this in the studio design.
- 3. Statistical Analysis, line 107: “Accuracy and precision were calculated.” Is reffering to what? Horizontal and vertical ? Please, specify
- Table 1: 7 patients with normal pressure hydrocephalus are included in the study, of which 5 are included in the 10-24 mmHg group. These patients do not have intracranial hypertension by definition. I ask the Authors if they have measured ICP> 20 mmHg in these patients. The case mix objectively brings together very different pictures of ICP modification: in the limitations the authors argue that this has an insignificant weight. I believe instead that the rapidity of increase of intracranial hypertension has a great influence on ONSD and in such a small sample, the inhomogeneity of patients becomes influential. I would modify the paragraph on Limitations and reformulate the statements in the Discussion / Conclusion / abstract
- Figure 1: in this figure, Authors can understand the reason for my question in point 8). It appears that the continuous variable (invasive ICP) was only compared with 25 measurements of US ONSD. Thus the values of the other two measurements made are lost. Since there are no assumptions of an anatomy difference between the two eyes, it would be easier to work with average values of horizontal / vertical and right / left, but to save the highest number of measurements performed with ultrasound. Please, specify
- Figure 3: in the notes on the figure is written ONDS, I think it is a mistake. Again, it seems that only 25 measurements were used for Bland Altman and not the 75 (25 x 3) taken. In the caption, the authors speak of a "small bias": how do they define it as small? How is the bias tolerance range calculated?
- 3. ROC Analysis, first paragraph: Is 5.1 the value in millimeters of ONSD corresponding to 10 mmHg of invasive ICP? It seems to me a value quite in contrast with the literature data that define ICP> 20 mmHg with ONSD between 5 and 6 mm. Please specify this ROC result well.
- Discussion, line 164: remove Optic Nerve Sheat Diameter and use acronysm
- Discussion, first paragraph: the authors state that there are no publications on ONSD measurement in awake patients with spontaneous breathing, when they themselves published a series of 15 patients in 2015 and other authors subsequently did so. Please review this statement
https://doi.org/10.1186/2197-425X-3-S1-A608
https://doi.org/10.1016/j.pediatrneurol.2015.08.009
doi:10.1001/jamaophthalmol.2017.6560
- Discussion, line 181: the Authors state citing only one work that US ONSD assessment has "high intra- and inter- observer reliability". It would be very nice if that were the case, but unfortunately we know that this, like most of the ultrasound methods, is very operator dependent. We are looking for new biomarkers / settings to standardize the procedure. Please review the statement.
- Discussion, fourth paragraph: in this passage of the discussion the concept of "increased ICP" on a value of 10 mmHg is inserted. Please specify how this value was chosen and what kind of clinical consideration is given to it.
- Conclusion: Conclusions should be less generic and better related to the novelty introduced by paper.
- References: unfortunately the references should be completely updated, as they all predate the period of the study (2015), which have therefore been partially overtaken by stronger papers, some of which I have already indicated in the review
Reviewer 2 Report
Although there are several non-invasive intracranial pressure measurements, the authors have shown significant association between optic nerve sheath diameter measurement via the eye and measured intracranial pressure in patients with hydrocephalus. More specifically, it was suggested that the intracranial pressure may be increased when the optic nerve sheath diameter exceeds 5.1 mm.
The following points need to be clarified.
1, There is no display of the image used for the actual measurement, and I do not know what resolution it is visible and what is being measured.
2, It is also necessary to describe whether or not measurement errors are likely to occur after eye surgery (for example, cataract). In addition, it is easy to understand if there are other cases that cannot be measured or are difficult to measure.
3, There are individual differences in humans, and their height and head size vary. Along with this, there may be individual differences in the optic nerve sheath diameter, but it is necessary to mention the relationship between the measured values and the individual differences.
4, There is no (v) in Introduction.
Reviewer 3 Report
Review of the manuscript titled „Real-Time Evaluation of Optic Nerve Sheath Diameter (ONSD) in Awake, Spontaneously Breathing Patients“.
Dr. Weidner and colleagues have assessed the diagnostic ability of measuring the optic nerve sheet diameter in assessment of intracranial pressure. The bedside non-invasive acquisition of intracranial pressure data would mean a possible reduction of surgical complications related to invasive ICP monitoring.
In regard to the manuscript we would like to address following concerns and comments:
Major comments
Some unclarity exist towards the sample of patients which is described. First in the limitations it is mentioned/clarified that this is a collective of hydrocephalus patients. Chronic mild elevated intracranial pressure as seen in idiopathic intracranial hypertension and its effects on the optic nerve vastly differs from patients with traumatic injury induced acutely elevated intracranial pressure. In the latter group, optic nerve involvement is rarely observed. All these pathologies do not “result in the same final course”. Therefore results in a cohort of chronic hydrocephalus cases cannot be readily extrapolated to all patients with raised intracranial pressure. The diagnosis leading up to invasive intracranial pressure monitoring, defining the collective of patients, should be provided in the materials and methods. For now, one can only assume these are patients suffering from idiopathic intracranial hypertension being evaluated for potential shunt placement. If however, different types of hydrocephalus were included, this information should be provided.
All statistics of continuous variables assume normality although no information is provided on verifying whether data is indeed normally distributed. If parametric testing is performed please provide information on normality testing (Shapiro-Wilk or Q-Q plotting).
The low dichotomization cut-off point (>10 mmHg) does not provide evidence of higher diagnostic precision as is suggested in the discussion, but reduces external validity for patients suffering from increased intracranial pressure (>25 mmHg). It remains - based on these data - uncertain whether the relation between ICP and ONSD remains linear in higher ICP ranks.
The accuracy and precision analyses are based on measurements comparing horizontal or vertical views not ICP data. The possible conclusion might be that only one section would suffice for ICP assessment. This does not however confirm accuracy and precision towards ICP assessment as is suggested in the discussion.
Sonographic diameters were assessed three times for every plane which results in six measurements per patient. How were these repeated measurements treated statistically? Was each measurement considered as an individual data point or meaned per individual?
How were ICP measurements timely synchronized with ONSD measurements. Or in other words, when (in relation to ONSD measurements) an how (single vs. multiple measurements) does the dichotomization into both intracranial pressure groups took place?
Minor comments
Were ultrasounds performed by a single assessor or multiple assessors? Where these assessors blinded to ICP measurements.
In figure 3, the x-axis is labeled as Mean ONDS, which should read Mean ONDS.
Please provide some insights in the pathophysiology optic nerve edema and damage as the result of increased intracranial pressure.
In general, these data provide evidence for the application of optic sheet ultrasonography as a non-invasive alternative to assess intracranial pressure in patients suffering from chronic hydrocephalus. These data can however not be simply extrapolated to all diagnoses leading up to raised intracranial pressure such as traumatic brain injury as is suggested by the introduction and discussion.